

# Phytochemical composition and in vitro antioxidant activities of *Citrus sinensis* peel extracts

Sok Sian Liew[1], Wan Yong Ho[1], Swee Keong Yeap[2] and Shaiful Adzni Bin Sharifudin[1,3]

[1] Department of Biomedical Sciences, The University of Nottingham Malaysia Campus, Semenyih, Selangor, Malaysia
[2] China-ASEAN College of Marine Sciences, Xiamen University Malaysia, Sepang, Selangor, Malaysia
[3] Biotechnology Research Centre, Malaysian Agricultural Research and Development Institute (MARDI), Serdang, Selangor, Malaysia

## ABSTRACT

**Background**. *Citrus sinensis* peels are usually discarded as wastes; however, they are rich sources of Vitamin C, fibre, and many nutrients, including phenolics and flavonoids which are also good antioxidant agents. This study aimed to examine phytochemical composition and antioxidant capabilities of *C. sinensis* peel extracted conventionally with different methanol/water, ethanol/water, and acetone/water solvents.

**Methods**. *C. sinensis* peels were subjected to extraction with 100%, 70% and 50% of methanol, ethanol, and acetone, respectively, as well as hot water extraction. Antioxidant activities of the peel extracts were examined via the 2,2-diphenylpicrylhydrazyl (DPPH) free radical scavenging activity, ferric reducing antioxidant power (FRAP) assay, and oxygen radical absorbance capacity (ORAC) assay. Total phenolic content and total flavonoid content of the extracts were measured via the Folin-Ciocalteau method and the aluminium chloride colorimetric method, respectively. Phenolic acid and organic acid composition of the peel extracts were further determined via high performance liquid chromatography (HPLC) while flavonoid content was identified via ultra performance liquid chromatography (UPLC).

**Results**. DPPH radical scavenging activity of *C. sinensis* peel extracts varied from 8.35 to 18.20 mg TE/g, FRAP ranged from 95.00 to 296.61 mmol Fe(II)/g, while ORAC value ranged from 0.31 to 0.92 mol TE/g. Significant level of association between the assays was observed especially between TPC and FRAP (R-square = 0.95, $P < 0.0001$). TPC of various *C. sinensis* peel extracts ranged from 12.08 to 38.24 mg GAE/g, with 70% acetone/water extract (AEC) showing the highest TPC. TFC ranged from 1.90 to 5.51 mg CE/g. Extraction yield ranged from 0.33 to 0.54 g/g DW and tended to increase with increasing water concentration in the solvent. In the phytochemical investigation, five phenolic acids were identified using HPLC, including gallic acid, protocatechuic acid, 4-hydroxybenzoic acid, caffeic acid and ferulic acid. A total of five organic acids including lactic acid, citric acid, L-mallic acid, kojic acid and ascorbic acid were quantified via HPLC. In addition, concentrations of six flavonoids including catechin, epigallocatechin, vitexin, rutin, luteolin and apigenin were determined via UPLC.

**Discussion and Conclusion**. Phytochemicals including phenolics and flavonoids in *C. sinensis* peel extracts exhibited good antioxidant properties. Among the extracts,

Corresponding author
Wan Yong Ho,
WanYong.Ho@nottingham.edu.my

70% AEC with highest TPC and high TFC content showed greatest antioxidant activity in all three assays. Different phenolic acids, organic acids and flavonoids were also identified from the extracts. This study indicated that *C. sinensis* peels contained potential antioxidant compounds which could be exploited as value added products in the food industry.

## INTRODUCTION

*Citrus sinensis*, also known as sweet orange, is consumed not only as a fruit but also as medicinal herb in some nations. It belongs to the Rutaceae family and is widely distributed in the tropical and subtropical regions. Annual worldwide citrus fruit production now stands at over 110 million tons, and oranges have become the most commonly grown fruit in the world (*Blauer, 2016*). Out of the forecasted global production of 24 million tons of oranges in 2016/17, it is estimated that around 8.3% (2 million tons) of these will be used for orange juice production (*Foreign Agricultural Service/USDA, 2017*). However, orange peels accounts for around 44% of the fruit body (*Li, Smith & Hossain, 2006*) and thus will produce a huge mass of by-products. These orange peels are usually discarded as waste, leading to serious disposal problem that may be detrimental to the environment.

Considering the huge quantity of "waste" that is produced in the food supply chain, orange peels offers a huge potential to be exploited as a value-added product, including for the recovery of natural antioxidants, pectin, enzymes or for the production of ethanol, organic acids, essential oils and prebiotics single cell protein (*Mamma & Christakopoulos, 2014*). In addition, the *C. sinensis* peel is a rich source of vitamin C, fibre, and many nutrients, including phenolics and flavonoids. It is subdivided into two main parts, epicarp and mesocarp. Epicarp is the coloured peripheral surface, largely made of parenchymatous cells and cuticle. It is covered with an epidermis of epicuticular wax with many small aromatic oil glands giving its particular smell. Mesocarp is the soft whitish middle layer lying beneath epicarp. It is made up of tubular-like cells connecting together to create the tissue mass compressed into the intercellular area (*Favela-Hernández et al., 2016*). The *C. sinensis* peel has been used as a traditional medicine in certain parts of the world for relieving stomach discomfort, skin inflammation, ringworm infections, aiding in neuroprotection, and improving heart health (*Li, Lo & Ho, 2006*; *Ghasemi, Ghasemi & Ebrahimzadeh, 2009*).

Various potent antioxidants have been found in citrus peels and showed antioxidant effects including free radical scavenging and metal chelation activities. It is encouraging to explore the active phytochemicals in *C. sinensis* peel, as reactive oxygen species play a main role in many diseases such as cancer, cardiovascular dysfunction, neurodegenerative diseases, and process of ageing (*Rafiq et al., 2016*). A recent study on identification of 4′-geranyloxyferulic (GOFA) among citrus peel extracts revealed that *C. sinensis* has the richest content of GOFA, which previously showed neuroprotective and dietary feeding colon cancer chemopreventive effects in rats (*Genovese & Epifano, 2012*; *Genovese et al.,*

*2014*). A group of flavonoids, polymethoxyflavones (PMFs), which is found abundantly and almost only from the citrus peels, have been given great attention because of their wide range of properties. Many *in vitro* experiments elucidated anticancer actions by PMFs such as antiproliferation, enzyme inhibition and cancer cell growth inhibition (*Qiu et al., 2011*; *Onda et al., 2013*; *Rawson, Ho & Li, 2014*).

Extraction is the key step for analysis and exploitation of plant bioactive compounds. An ideal extraction method should be quantitative, non-destructive, and time effective. Due to lower toxicity and ease of access to water, the traditional method of using medicinal plants was by boiling them in water and consuming the extracts either as additives in food or directly as functional foods, but the effectiveness of consuming such boiled water extract was in doubt (*Wong et al., 2006*). Conventional solvent extraction (CSE) is widely used for the recovery of bioactive compounds due to its simplicity, despite some disadvantages such as long extraction time, large consumption of solvents, exposure to flammable and hazardous liquid organic solvents. However, TPC and antioxidant activities of extracts extracted via CSE were comparable to those via other non-conventional methods as shown in a study by *Nayak et al. (2015)*. Phenolics or antioxidant content is greatly affected by properties of the extracting solvents. Some common solvent used in the extraction of phenolics are methanol, ethanol, propanol, acetone and ethyl acetate (*Spigno, Tramelli & De Faveri, 2007*). Phenolic compounds dissolve better in solvent with a higher polarity such as methanol. It is important to note that some organic solvents are identified as toxic such as methanol; therefore, ethanol as a food-grade solvent is preferred to be used for the extraction of phenolic compounds from various citrus peels (*Li, Smith & Hossain, 2006*).

Due to their low cost and high availability in the world, *C. sinensis* peels and their phytochemical compounds could serve as a cheap and yet nutritional dietary supplement or even as a potential therapeutic agent. However, the health benefits of *C. sinensis* should be proven with a more reliable and systematic study. It was hypothesized that conventional extraction with the solvents including methanol, ethanol, and acetone can isolate the useful bioactive compounds which will exert high antioxidant activities. This study aims to examine antioxidant capabilities of *C. sinensis* peel extracts and the correlation to their phytochemical content.

## MATERIALS & METHODS

### Extraction and preparation of samples

*C. sinensis* peels were by-products collected from an orange juice manufacturer. The peels composed of flavedo and albedo were then washed and fully dried in an oven at 60 °C for 72 h. The dried peels were ground to powder with particle size ranging of 0.5 mm to 0.1 mm using mortar and pestle and were extracted using 100/0, 70/30, 50/50% (v/v) of methanol/water, ethanol/water, and acetone/water solvents respectively for 72 h with a mass to volume ratio of 1:25 (g/ml). The extracts were then filtered through Whatman No. 1 filter paper and collected into glass bottles. The whole process of extraction and filtration was repeated twice followed by evaporation of the collected extracts to dryness using a rotary evaporator at 37 °C. The extracts were re-dissolved in dimethyl sulphoxide

(DMSO) to a concentration of 100 mg/mL and kept at 4 °C until use. Extraction yield was expressed in g of extract per g of dry weight (g/g DW). For the water extract, 4 g of dried *C. sinensis* peel powder was boiled with distilled water for 1 h before filtering it through Whatman No. 1 filter paper. The whole process of extraction and collection of extract was repeated followed by evaporating the collected extract to 40 mL (Concentration 100 mg/mL) using a boiling water bath. Water extract stock was stored in −20 °C and thawed before use. Extracts of *C. sinensis* peel by using 100, 70, 50 wt.% methanol/water and ethanol/water solvents are annotated as 100% MEC, 70% MEC, 50% MEC, 100% EEC, 70% EEC, and 50% EEC, respectively. Extracts of *C. sinensis* peel by using 70 and 50 wt.% acetone/water solvents are annotated as 70% AEC, and 50% AEC respectively. Water extract of *C. sinensis* peel are annotated as WEC.

### Determination of 2,2-diphenyl-1-picrylhydrazyl (DPPH) radical scavenging activity

The scavenging activity of the extract against DPPH ● radical was measured according to *Brand-Williams, Cuvelier & Berset (1995)*. A total of 50 µL of 2.5 mg/mL extract was added to 150 µL of mixture (0.2 mM DPPH ● and 90 mM Tris-Cl). The total volume was made up to 200 µL with methanol. The mixture was incubated in dark for 40 min at 25 °C before taking the absorbance readings at 517 nm. DPPH ● scavenging radical ability of each sample was expressed as mg of trolox equivalents per g of sample (mg TE/g).

### Determination of ferric reducing antioxidant power (FRAP)

FRAP assay was performed with slight modification (*Benzie & Strain, 1996*). The FRAP reagent was prepared by mixing TPTZ (1 mM), $FeCl_3 \bullet 6H_2O$ (2 mM), and 300 mM acetate buffer in a ratio of 10:1:1 at 37 °C. 25 µL of 1 mg/mL extract was mixed with 175 µL FRAP reagent. A final volume of 200 µL reaction mixture was incubated in dark for 10 min at 25 °C before taking the absorbance readings at 590 nm. $FeSO_4 \bullet 7H_2O$ with different concentrations (100–1,000 µM) was used as standard for construction of calibration curve. FRAP value of each sample was expressed as mmol of Fe(II) per g of sample (mmol Fe(II)/g).

### Determination of oxygen radical absorbance capacity (ORAC)

The ORAC assay was carried out according to the method of *Huang et al. (2002)* with modifications. First, 50 µL of 100 µg/mL extract was added to 800 nM fluorescein in 75 mM phosphate buffer pH 7.4. The reaction mixture was incubated at 37 °C for 15 min followed by the addition of 200 mM AAPH (2,2′-azobis-2- methyl-propanimidamide, dihydrochloride) solution to a final volume of 200 µL. The fluorescence signal was measured using a Hitachi F-7000 Fluorescence Spectrophotometer in 5 min interval over 90 min by excitation at 485 nm, emission at 520 nm. Trolox was used as standard for construction of calibration curve.

The area under the curve (AUC) of each sample was calculated by integrating the relative fluorescence curve. Next, net AUC of the sample was calculated by subtracting the AUC of the blank from the AUC of the sample.

$$AUC = 1 + RFU1/RFU0 + RFU2/RFU0 + \cdots\cdots + RFU18/RFU0$$

where RFU0, relative fluorescence value of time point zero; RFUx, relative fluorescence value for the number of reading (e.g., RFU5 is relative fluorescence value of fifth reading, which is at minute 25)

$$\text{Net } AUC = AUC\ (\text{sample}) - AUC\ (\text{blank}).$$

The regression equation between net AUC and Trolox concentrations was determined and the ORAC value of extracts was expressed as mol Trolox equivalents per gram of sample (mol TE/g).

### Determination of total phenolic content

TPC of the extracts was determined according to the Folin–Ciocalteu method (*Singleton & Rossi, 1965*). A total of 250 µL of 2 N Folin–Ciocalteu reagent was mixed with 50 µL of 10 mg/mL extract, following by addition of 750 µL of 7% w/v Na2CO3 after 5 min. The total volume was made up to 5 mL with distilled water. The mixture was incubated in dark for 2 h at 25 °C before absorbance was measured using a spectrophotometer (The VersaMax$^{TM}$ Microplate Reader; Molecular Devices, San Jose, CA, USA) at 765 nm. TPC results were expressed as mg gallic acid equivalents per g of dry weight (mg GAE/g DW).

### Determination of total flavonoid content

TFC of extracts was determined using the aluminium chloride colorimetric assay with slight modification (*Zhishen, Mengcheng & Jianming, 1999*). A total fo 25 µL of 10 mg/mL extract was added with 7.5 µL of NaNO2 (5% w/v), and 7.5 µL of AlCl3 (10% w/v). The mixture was then allowed to stand for 10 min at 25 °C. 50 µL of NaOH (1 M) was added subsequently and the total volume was made up to 250 µL with distilled water. The absorbance was measured at 510 nm using a spectrophotomoter. TFC results were expressed as mg of catechin equivalents per g of dry weight (mg CE/g DW).

### Quantification of phenolic acids content

Identification and quantification of phenolic acids in the orange peel extracts was performed using a High Performance Liquid Chromatography (HPLC) method using 2956 LC system (Waters, USA). Samples were filtered through 0.22 µm pore size membrane filters before injection. The presence of phenolic acids was then determined using a reversed phase XBridge C18 column (4.6 × 100 mm, 3.5 µm particle size) and the detector was set at $\lambda = 270$ nm, and $\lambda = 306$ nm. The separation of phenolic acids was made in gradient condition at 30 °C, using a mobile phase A made of acid water (0.1% formic acid) and mobile phase B, methanol (100%) with the flow rate of 0.7 mL/min. The gradient elution was performed as follows: 0–10 min, from 95% to 85% A; 10–20 min, from 85% to 80% A; 20–52 min, from 80 to 70% A; 52–55 min, maintained at 70% A; 55–58 min, from 70 to 50% A; 58–63 min, from 50 to 20% A; 63–70 min, from 20 to 95% A; 70–75 min, maintained at 95% A. Peak identification was made by comparing retention time of known phenolic acids and quantification was performed using calibration curves obtained by injecting known amounts of the pure phenolic acids (gallic acid, vanillic acid, protocatechuic acid, syringic

acid, 4-hydroxybenzoic acid, caffeic acid, o-coumaric acid, ferulic acid, sinapic acid and p-coumaric acid) as the external standards.

## Determination of organic acids content

HPLC analyses of organic acid content was carried out using 2695 Alliance Separation Module (Waters, Milford, MA, USA) equipped with a 2996 diode array detector (Waters, Milford, MA, USA). A 10 $\mu$L aliquot of filtered sample was separated using Synergi Hydro-RP80A column (250 × 4.6 mm, 4 $\mu$m particle size) (Phenomenex, Torrance, CA, USA) with temperature controlled at 30 °C. The mobile phase consisted of mobile phase A (20 mM KH2PO4 with adjusted pH 2.9) and mobile phase B (water) with a flow rate of 0.6 mL/min. Gradient elution was performed as follows: 0–30 min, maintained at 100% A; 30–31 min, from 100% to 0% A; 31–45 min, maintained at 0% A; 45–46 min, from 0 to 100% A; 46–55 min, maintained at 100%. Peak identification was made by comparing retention times and UV spectra at 190, 210 and 254 nm with authentic organic acids compounds. Quantification was performed using calibration curves obtained by injecting known amounts of pure organic acids (tartaric acid, lactic acid, acetic acid, citric acid, succinic acid, oxalic acid, L-mallic acid, kojic acid and ascorbic acid) as external standards.

## Determination of flavonoids content using UPLC

The flavonoid content of the filtered extracts were separated using AcquityTM Ultra Performance Liquid Chromatography (UPLC) system (Waters, Milford, MA, USA) with Kinetex C18 100 A column (100 mm × 2.1 mm; 1.7 $\mu$m particle size), at a flow rate of 0.4 ml/min with the temperature controlled at 40 °C under the UV spectrum of 280, 330, 360 nm. The gradient elution consists of mobile phase A (water:acetic acid, 97:3) and mobile phase B (methanol). The gradient elution was conducted as follows; 0–1 min, maintained at 100% A; 1–10 min, from 100 to 40% A; 0–12 min, from 40 to 100% A and then maintained at 100% A for another 2 min. Quantification was performed using calibration curves obtained by injecting known amounts of flavonoids (Epigallocatechin, vitexin, rutin, quercetin, luteolin, apigenin, tannic acid and ellagic acid) as external standards with known retention time.

## Statistical analysis

All experiments were performed in triplicates unless stated otherwise. Statistical analyses of the experimental data were performed with GraphPad prism 6 statistical software (GrapPad Software, San Diego, CA, USA). Results of the replicates were expressed as mean ± standard error (SEM). One-way analysis of variance (ANOVA) with Tukey's multiple comparisons test was used to evaluate differences between means in each experiment. Experimental results were further analyzed for Pearson correlation coefficient (R-square) between TPC, TFC and different antioxidant assays. $P$ value of $\leq 0.05$ was taken as statistically significant.

# RESULTS

## Antioxidant activity

Antioxidant activities of the extracts were evaluated via 2,2-diphenyl-1-picrylhydrazyl (DPPH) radical scavenging activity, ferric reducing antioxidant power (FRAP), and oxygen

**Table 1** Antioxidant activities of *C. sinensis* peel extracts.

| Sample | DPPH (mg TE/g) | FRAP (mmol Fe(II)/g) | ORAC (mol TE/g) |
|---|---|---|---|
| 100% MEC | $13.96 \pm 1.08^{abc}$ | $275.62 \pm 1.85^{ab}$ | $0.73 \pm 0.05^{a}$ |
| 70% MEC | $16.69 \pm 1.20^{ab}$ | $240.94 \pm 4.95^{c}$ | $0.70 \pm 0.04^{ab}$ |
| 50% MEC | $15.98 \pm 1.33^{ab}$ | $214.64 \pm 3.49^{d}$ | $0.56 \pm 0.03^{bcde}$ |
| 100% EEC | $11.61 \pm 0.82^{ac}$ | $139.94 \pm 3.89^{e}$ | $0.46 \pm 0.03^{cef}$ |
| 70% EEC | $16.52 \pm 1.29^{ab}$ | $219.02 \pm 3.87^{cd}$ | $0.68 \pm 0.02^{ad}$ |
| 50% EEC | $15.96 \pm 1.32^{ab}$ | $194.73 \pm 5.81^{d}$ | $0.50 \pm 0.02^{cg}$ |
| 70% AEC | $18.20 \pm 1.62^{b}$ | $296.61 \pm 7.97^{a}$ | $0.92 \pm 0.03^{h}$ |
| 50% AEC | $16.87 \pm 1.30^{ab}$ | $269.71 \pm 7.33^{b}$ | $0.60 \pm 0.02^{aeg}$ |
| WEC | $8.35 \pm 1.14^{c}$ | $95.00 \pm 2.11^{f}$ | $0.31 \pm 0.03^{f}$ |
| Ascorbic acid | $1883.97 \pm 22.09$ | $14672.83 \pm 218.86$ | $5.33 \pm 0.69$ |
| Gallic acid | $4133.73 \pm 360.07$ | $26059.73 \pm 4427.54$ | $7.88 \pm 0.60$ |

Notes.

[abcdefgh] Mean $\pm$ SEM followed by different alphabets in the same column were significantly different between the *C. sinensis* peel extracts at $P < 0.05$ by one-way ANOVA.

TE, trolox equivalents; Fe(II), amount of $Fe^{2+}$ reduced from $Fe^{2+}$.

Extracts of *C. sinensis* peel by using 100, 70, 50 wt.% methanol/water and ethanol/water are annotated as 100% MEC, 70% MEC, 50% MEC, 100% EEC, 70% EEC, and 50% EEC, respectively. Extracts of *C. sinensis* peel by using 70 and 50 wt.% acetone/water solvents are annotated as 70% AEC, and 50% AEC respectively. Water extract of *C. sinensis* peel are annotated as WEC.

**Table 2** Correlation between antioxidant activities and phytoconstituents of *C. sinensis* peel extracts.

| Correlation R-square | DPPH | FRAP | ORAC | TFC |
|---|---|---|---|---|
| TPC | $0.83^{a}$ | $0.95^{a}$ | $0.80^{a}$ | $0.91^{a}$ |
| TFC | $0.76^{a}$ | $0.93^{a}$ | $0.66^{a}$ | |
| ORAC | $0.61^{a}$ | $0.82^{a}$ | | |
| FRAP | $0.72^{a}$ | | | |

Notes.

[a] Correlation of the experimental values between the tests were statistical significant at $P < 0.05$.

radical absorbance capacity (ORAC). DPPH radical scavenging activity of *C. sinensis* peel extracts varied from 8.35 to 18.20 mg TE/g, FRAP ranged from 95.00 to 296.61 mmol Fe(II)/g, while ORAC ranged from 0.31 to 0.92 mol TE/g. In all three assays, 70% AEC showed higher antioxidant activity and WEC showed a much lower antioxidant activity among the extracts (Table 1). However, DPPH, FRAP and ORAC values of extracts were much lower than the tested positive controls, ascorbic acid and gallic acid. Pearson correlation coefficients (R-square) between TPC, TFC and different antioxidant assays were tabulated in Table 2. Significant level of association between the assays was observed especially between FRAP values and TPC (R-square = 0.95, $P < 0.0001$) as well as TFC (R-square = 0.93, $P < 0.0001$) across all the extracts.

## Extraction yield, total phenolic and total flavonoid content

In general, extraction yield of all samples ranged from 0.33 to 0.54 g/g DW and appeared to increase with increasing water concentration in the solvent (Table 3). 100% AEC was excluded from further study due to its low yield (<0.005 g/g DW) and low solubility (Table 3). TPC of various *C. sinensis* peel extracts ranged from 12.08 to 38.24 mg GAE/g, with 70%

**Table 3** Extraction yield, total phenolic and flavonoid content of *C. sinensis* peel extracts.

| Sample | Extraction yield (g/g DW) | TPC (mg GAE/g) | TFC (mg CE/g) |
|---|---|---|---|
| 100% MEC | 0.41 | 36.09 ± 2.87[a] | 4.61 ± 0.08[abc] |
| 70% MEC | 0.47 | 34.55 ± 2.09[a] | 4.32 ± 0.19[ab] |
| 50% MEC | 0.51 | 29.48 ± 2.49[ab] | 3.81 ± 0.13[a] |
| 100% EEC | 0.33 | 21.38 ± 0.93[bc] | 2.59 ± 0.09[e] |
| 70% EEC | 0.48 | 33.07 ± 2.66[ab] | 4.35 ± 0.20[af] |
| 50% EEC | 0.52 | 29.65 ± 2.25[ab] | 3.73 ± 0.22[a] |
| 100% AEC | 0.00 | NA | NA |
| 70% AEC | 0.52 | 38.24 ± 3.44[a] | 5.03 ± 0.27[bdf] |
| 50% AEC | 0.54 | 35.58 ± 2.81[a] | 5.51 ± 0.43[cd] |
| WEC | – | 12.08 ± 0.96[c] | 1.90 ± 0.09[e] |

**Notes.**

[abcdef] Mean ± SEM followed by different alphabets in the same column were significantly different at $P < 0.05$ by one-way ANOVA.

DW, dry weight; GAE, gallic acid equivalents; CE, catechin equivalents.

Extracts of *C. sinensis* peel by using 100, 70, 50 wt.% methanol/water and ethanol/water are annotated as 100% MEC, 70% MEC, 50% MEC, 100% EEC, 70% EEC, and 50% EEC, respectively. Extracts of *C. sinensis* peel using 100, 70 and 50 wt.% acetone/water solvents are annotated as 100% AEC, 70% AEC, and 50% AEC respectively. Water extract of *C. sinensis* peel are annotated as WEC. NA, Not available (The yield of 100% acetone was lower than 0.005 g/g DW and was not soluble, thus not able to be used for subsequent analysis.)

AEC showing the highest TPC. The other extracts exhibited relatively high TPC too except for 100% EEC and WEC which showed significantly lower TPC than the other extracts. On the other hand, TFC ranged from 1.90 to 5.51 mg CE/g. 50% AEC showed the highest TFC, followed by 70% AEC. Generally, aqueous acetone extracts contained higher TPC and TFC than the other extracts while 100% EEC and WEC contained the lowest TPC and TFC among all.

## Phytochemical analysis

Phenolic acids can be divided into derivatives of benzoic acid and of cinnamic acid and both derivatives were found in our peel extracts. Cinnamic acid derivatives namely ferulic acid and caffeic acid were found in highest abundance while the derivatives of benzoic acid such as gallic acid, protocatechuic acid and 4-hydroxybenzoic acid were present at lower abundance as compared to the former group (Table 4).

Flavonoids are classified into six groups including flavanone, flavonol, flavone, isoflavone, flavan-3-ols, and anthocyanin. The major class of flavonoids in the extracts appear to be the flavan-3-ols (catechin and epigallocatechin), followed by flavanone (luteolin, apigenin and vitexin) while flavonol (rutin) was present at low abundance in the extracts (Table 4).

On the other hand, a few organic acids were identified in the extracts via HPLC (Table 4). Interestingly, 100% MEC, 100% EEC and 70% EEC were shown to contain only lactic acid. Citric acid is the major organic acid in the remaining extracts, followed by lactic acid and L-mallic acid. Kojic acid and ascorbic acid appeared to be present in much lower abundance in the extracts.
**Table 4 Phytochemical content of *C. sinensis* peel extracts.**

| Phytochemicals (ug/g extract) | 100% MEC | 70% MEC | 50% MEC | 100% EEC | 70% EEC | 50% EEC | 70% AEC | 50% AEC | WEC |
|---|---|---|---|---|---|---|---|---|---|
| **Phenolic acids** | | | | | | | | | |
| Gallic acid | 43.43 | 18.80 | 32.69 | 31.30 | 42.67 | 38.88 | 33.55 | 40.14 | 20.14 |
| Protocatechuic acid | 59.87 | 140.25 | 108.95 | 70.33 | 121.25 | 133.26 | 112.08 | 130.79 | 24.40 |
| 4-hydroxybenzoic acid | 65.22 | 53.93 | 69.79 | 63.48 | 73.05 | 81.74 | 50.51 | 54.25 | 24.07 |
| Caffeic acid | 224.65 | 247.96 | 411.75 | 164.99 | 243.01 | 362.70 | 266.43 | 264.84 | 69.58 |
| Ferulic acid | 377.61 | 821.87 | 769.19 | 579.33 | 404.34 | 742.22 | 917.88 | 683.44 | 108.79 |
| **Organic acids** | | | | | | | | | |
| Lactic acid | 18660.69 | 20981.33 | 16387.22 | 20190.94 | 16316.47 | 12708.52 | 12444.50 | 10117.39 | 20929.25 |
| Citric acid | 0 | 40900.07 | 38405.91 | 0 | 0 | 42559.18 | 53673.85 | 45076.92 | 37364.21 |
| L-mallic acid | 0 | 9207.74 | 9200.43 | 0 | 0 | 8701.07 | 5876.59 | 9378.31 | 1308.75 |
| Kojic acid | 0 | 247.64 | 218.13 | 0 | 0 | 222.96 | 221.39 | 229.17 | 120.68 |
| Ascorbic acid | 0 | 12.55 | 2.19 | 0 | 0 | 4.37 | 9.71 | 13.55 | 6.21 |
| **Flavonoids** | | | | | | | | | |
| Catechin | 248.76 | 514.63 | 627.22 | 123.99 | 573.81 | 669.62 | 572.26 | 679.32 | 352.72 |
| Epigallocatechin | 373.41 | 472.66 | 621.84 | 255.24 | 436.16 | 593.99 | 446.57 | 520.82 | 178.07 |
| Vitexin | 103.92 | 190.17 | 196.80 | 93.12 | 107.60 | 202.40 | 225.52 | 195.38 | 69.61 |
| Rutin | 23.48 | 20.34 | 20.38 | 30.63 | 29.48 | 22.36 | 21.63 | 26.63 | 15.08 |
| Luteolin | 227.97 | 496.83 | 539.49 | 334.51 | 300.51 | 458.37 | 516.27 | 478.70 | 324.74 |
| Apigenin | 141.24 | 285.77 | 313.08 | 203.65 | 174.91 | 270.45 | 311.19 | 279.59 | 194.38 |

**Notes.**

Each data point indicates the average results of phytochemicals phenolic acid, organic acid and flavonoids in ug/g extract of *C. sinensis* peel extracts.

## DISCUSSION

In this study, antioxidant activities of *C. sinensis* peel extracts were evaluated and correlated with the important phytochemical content including phenolic acid, flavonoid and organic acid. Antioxidants can deactivate radicals via two main mechanisms, hydrogen atom transfer (HAT) and single electron transfer (SET). In HAT, antioxidant donates hydrogen atoms to stabilise free-radical species to quench them from progressing further in radical reactions while in SET, free radicals are reduced through the donation of an electron from antioxidant compounds (*Craft et al., 2012*). Depending on the structure and properties of the antioxidants present, either HAT or SET may dominate in a given system (*Prior, Wu & Schaich, 2005*). Therefore, antioxidant capacities of plant extracts greatly depends on extract composition as well as conditions and mechanism of the test used. In order to evaluate antioxidant activity of components in the extract, three antioxidant assays operated on different mechanism were used in our study; FRAP and ORAC assay measures via HAT and SET respectively while DPPH assay determines antioxidant activity via both mechanism (*Prior, Wu & Schaich, 2005*). In both ethanol and acetone extracts, greatest antioxidant activity was observed in 70% followed by 50% and 100% of both extracts in all three assays (Table 1). This suggests that water content in the extracts may not correlate

proportionally to the antioxidant level but addition of water to the extraction could improve the antioxidant level of the ethanol and acetone extracts. In contrast, antioxidant level of the methanol extracts did not correlate well to water content in the extracts. 100% MEC showed antioxidant activity that was superior to 70% and 50% MEC in both FRAP and ORAC assays but the reverse in DPPH assay.

Antioxidant capacity of phenolics and flavonoids in plants is the main contributor to the specific biological actions in diseases prevention and treatment (*Dai & Mumper, 2010*). Therefore, bioactive phytochemical components may define the medicinal value of a plant source. From the results, we observed a high correlation between FRAP values and TPC and TFC across all the extracts (Table 2). The higher correlation of TPC and TFC to FRAP values suggest to us that the antioxidant secondary metabolites, in particular phenolics and flavonoids in the peel extracts, may react with free radicals mainly via SET mechanism. Overall, statistically significant correlation between TPC, TFC and the antioxidant assays suggests that the phenolic and flavonoid content contribute to antioxidant activities of the *C. sinensis* extracts.

Phenolics or antioxidant content is greatly affected by properties of the extracting solvents (*Spigno, Tramelli & De Faveri, 2007*). Phenolic compounds are generally known to dissolve better in solvents with higher polarity. Polar alcohol type solvent would produce higher yield as compared to other type of solvents. Extraction yield may be increased with addition of water to ethanol, but water content in the solvent would increase concomitant extraction of other compounds, yielding lower concentration of phenols in the extracts (*Naczk & Shahidi, 2006*). In agreement to the former, we found that the yield of extract increased with increasing water percentage within each solvent extraction group. Although there isn't a specific pattern observed with regards to the effect of addition of water to TPC or TFC content, 30% of water content in each of the solvent extraction group generally exhibited higher TPC and TFC content than those with 50% of water content (Table 3). Water at boiling temperature was claimed to be one of the effective solvents for antioxidants extraction giving higher total phenol content (*Sousa et al., 2008*). In our study, WEC had exerted antioxidant activities and tested to have phenolic and flavonoid content but being the least among the extracts (Tables 1 & 3). It was understood that conventional solvent extraction (CSE) is generally being used to extract bioactive components from the plant materials at a small scale level. Our study which used CSE served as a control method to understand the bioactivities of *C. sinensis* peels and a reference method for small scale production or homemade level. The major hurdle to scale up the extraction with this conventional method would be long extraction time and large consumption of solvents. Therefore, new and promising non-conventional extraction techniques were introduced for industrial application such as ultrasound-assisted extraction (UAE), molecular distillation, microwave-assisted extraction (MAE), pulsed electric field extraction, and supercritical fluid extraction (*Selvamuthukumaran & Shi, 2017*). A recent study comparing CSE, UAE, MAE and supercritical $CO_2$ extraction of Maltease citrus peel showed that MAE was a more effective method in phenols and flavonoids extraction while CSE gave an extract with more antioxidant activity (*Boudhrioua, 2016*). In another study by *Ko, Kwon & Chung (2016)*, a pilot-scale subcritical water extraction plant was conducted to extract

antioxidant flavonoids from dried satsuma mandarin peel (*Citrus unshiu* Markovich) and the proportion of flavonoids recovered with this extraction pilot plant was 96.3%.

In addition to the important role of extraction solvent, pre-extraction and extraction conditions and methods are equally important in extracting compounds from plant materials. A study by *Hegde, Agrawal & Gupta (2015)* recommended peel drying since loss of water content from the peel decreases the bulk of the material for easier handling and storage, lower risk of bacterial growth, as well as more efficient extraction. The study concluded that extraction of sundried peel with acidified aqueous methanol at 90 °C for 5 h yielded the highest polyphenol content. However, it is understood that conditions of sun drying is not controlled throughout the process, and oven drying is the alternative option. Oven drying uses thermal energy to remove moisture from the samples rapidly and at the same time, preserves the phytochemicals. Grinding of samples into smaller particle size increases surface contact between samples and extraction solvents (*Azwanida, 2015*; *Hegde, Agrawal & Gupta, 2015*). These findings supported the use of oven drying at 60 °C following by grinding for pre-extraction method with *C. sinensis* peels in this study.

In this study, we further identified the different phenolic acids, flavonoids and organic acids content in the various extracts. Phenolic compounds are secondary metabolites synthesized by plants for protection against excessive ultraviolet radiation or pathogenic aggression (*Beckman, 2000*). Their biological benefits especially antioxidant properties have been extensively studied and described in the literature. Five phenolic acids were identified from *C. sinensis* peel extracts including gallic acid, protocatechuic acid, 4-hydroxybenzoic acid, caffeic acid and ferulic acid. Ferulic acid is the most abundant phenolic acid of *C. sinensis* peel in our study, in agreement with the previous study by *M'hiri et al., (2017)*. Ferulic acid has been reported with various bioactivities including antioxidant, anti-diabetic, anti-tumor and cardio-protection (*Kumar & Pruthi, 2014*). Similar to ferulic acid, caffeic acid which was present as the second most abundant phenolic acid in all the extracts, are hydroxycinnamic acid derivatives which were shown to show concentration-dependent antioxidant effects. These compounds exhibited inhibition against induced lipid peroxidation in mouse liver microsomes and scavenging activity against a range of radicals including nitric oxide, superoxide and 2,2′-azino-bis-3-ethylbenzthiazoline-6-sulfonic acid radical (ABTS+) (*Maurya & Devasagayam, 2010*). In addition, both ferulic acid and caffeic acid are widely added as active ingredients in cosmetic product due to its anti-aging, anti-hyaluronidase and UV absorption capacities (*Kumar & Pruthi, 2014*; *Taofiq et al., 2017*).

In addition, our results show that *C. sinensis* peel extracts contain flavonoids including catechin, epigallocatechin, vitexin, rutin, luteolin and apigenin. The addition of water to the methanol and ethanol extraction appeared to enhance the concentration of most flavonoids such as catechin, epillocatechin, vitexin and luteolin (Table 4). All extracts with exception of 100% MEC and 100% EEC contained catechin as the most abundant flavonoid. Catechin is found abundantly in tea extracts and is well known for its multiple health benefits including anti-aging, anti-diabetic and anti-cancer effects (*Pandey & Rizvi, 2009*). Epigallocatechin and apigenin which were also present in high abundance in the extracts were suggested to be able to reverse epigenetic changes in disease prevention and regulate a number of

biological processes (*Li et al., 2016*; *Shankar et al., 2016*; *Zhou, Yang & Kong, 2017*). On the other hand, rutin was claimed to demonstrate beneficial biological properties including antioxidant, anti-inflammatory and anticarcinogenic properties (*Rawson, Ho & Li, 2014*).

Apart from the polyphenols, the concentration of organic acids were also determined via HPLC method (Table 4). In comparison to the other phytochemicals, lactic acid, citric acid and L-malic acid were present in much higher concentration in all the extracts apart from 100% MEC, 100% EEC and 70% EEC. These organic acids are commonly found in citrus food and carry major economy value as they have been widely used as acidulant, preservative, emulsifier, flavorant and buffering agents across many industries particularly in food, beverage, pharmaceutical, nutraceutical and cosmetic manufacturing (*Ciriminna et al., 2017*). For example, ascorbic acid and citric acid are normally added to fruit beverages, as acidulant, to enrich the nutrient content and palatability of juices from orange, grapefruit and lemon (*Scherer et al., 2012*). In addition, the presence of ascorbic acid, limonoids citric acid and flavonoids content of *C. sinensis* had been claimed to be useful for fermentation and for treating kidney stones in clinical application (*Alok et al., 2014*). As global supply of organic acid particularly citric acid has rose from less than 0.5 to more than 2 million tonnes for the last two decades, extracting organic acid from citrus industrial waste can serve as alternative source that supplying the market needs of natural organic acids (*Ciriminna et al., 2017*).

## CONCLUSIONS

The antioxidant activity, total phenolic and flavonoid content of the *C. sinensis* peels were evaluated. The *C. sinensis* peels showed high antioxidant activities, total phenolic and flavonoid content. Conventional solvent extraction in our study has produced extracts with high antioxidant activities and high phytochemicals content. In particular, extraction of *C. sinensis* peels with 70 wt.% acetone/water solvent was found to be most effective in extracting organic acid (citric acid and lactic acid) and phenolic acid (ferrulic acid and caffeic acid) of the *C. sinensis* peel extracts. However, for scaled up industrial production, more effective technology can be considered such as MAE and supercritical $CO_2$ extraction.

The rich phytochemical constituents including phenolic and flavonoid content appeared to contribute to the antioxidant potential to the *C. sinensis* peel extracts. The bioactive phytochemicals could therefore be exploited for various applications such as for extraction of natural antioxidants, food additive and colourants in the food industry. Despite being agricultural wastes produced in the food supply chain, the enormous availability of *C. sinensis* peels could be benefited as value added products in line with green technology.

### Funding

This work was supported by University of Nottingham Malaysia Campus internal funding. The funders had no role in study design, data collection and analysis, decision to publish, or preparation of the manuscript.

## Grant Disclosures
The following grant information was disclosed by the authors:
University of Nottingham Malaysia Campus internal funding.

## Competing Interests
The authors declare there are no competing interests.

## Author Contributions
- Sok Sian Liew performed the experiments, analyzed the data, prepared figures and/or tables, authored or reviewed drafts of the paper, approved the final draft.
- Wan Yong Ho conceived and designed the experiments, performed the experiments, analyzed the data, contributed reagents/materials/analysis tools, prepared figures and/or tables, authored or reviewed drafts of the paper, approved the final draft.
- Swee Keong Yeap conceived and designed the experiments, analyzed the data, contributed reagents/materials/analysis tools, authored or reviewed drafts of the paper, approved the final draft.
- Shaiful Adzni Bin Sharifudin performed the experiments, analyzed the data, approved the final draft.

## Data Availability
   The raw data are provided in the Supplemental Files.

## Supplemental Information
Supplemental information for this article can be found online at http://dx.doi.org/10.7717/peerj.5331#supplemental-information.

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
