# Peer review of "Phytochemical composition and in vitro antioxidant activities of *Citrus sinensis* peel extracts"

_PeerJ, doi:10.7717/peerj.5331_

## Round 0.1 · original submission · Minor Revisions

You will see that, while the reviewers find your work of interest, they have raised a number of points that need to be addressed before we can make a decision on publication.

Reviewer 1 ·

Basic reporting

no comment

Experimental design

Material and Methods

1. I believe as the peels were collected from an orange juice manufacturer the waste is composed by the flavedo and albedo. I encourage the authors to clarify it in the M&M section.

2. How long the drying process took? I also encourage the authors to add the information about the drying time and size of particles.

3. Why choose water extraction process that use boil temperature since higher temperature for a long time may “destroy” phenolic compounds?

Validity of the findings

Conclusion

4. First paragraph of the conclusion seems to fit in the results section. Conclusion should relate directly to the questions brought up in the introduction. The second paragraph is a good example of it.

Additional comments

The current manuscript has significant information for the scientific society. Fruits and vegetable waste management is a challenge and also represent a company cost. Reduction, reuse and other strategies can be applied. Also, these wastes are rich in antioxidant components which can be used in the food industry. All the assays are appropriate for the proposal of the study. The manuscript requires a few adjustments to be more informative to reader before acceptance.

Reviewer 2 ·

Basic reporting

In general, the manuscript is written clearly and based on recent citations.

Considering that one of the objectives of the study was to analyze the phytochemical composition and antioxidant capacity of the C. sinensis peel by comparison of 3 extraction methods (L 4-6), it is strongly recommended that the introduction of the manuscript consider the different methodologies of extraction of these compounds and not only presents them in the Materials & Methods.

Citations that justify method choices are important. Issues such as: Advantages and disadvantages, possibility of changes in molecules due to the use of solvents, residual solvent, should be considered.

L 67-69: The authors mention "Many in vitro experiments ...", but only present a reference (Rawson, Ho and Li, 2014). Cite more references to substantiate the argument.

Experimental design

There is no bibliographical reference of methodologies for the preparation and extraction of the samples, it is fundamental that they provide them.

Validity of the findings

The practical investigation of the possible industrial applications of these extracted compounds is more innovative and of greater impact than the analysis of the compounds themselves. The latter seems to me very widespread in the literature.

Additional comments

As mentioned earlier, I believe that studies demonstrating the potential of compounds extracted in the industry are more relevant today. However, the investigation in question can be considered relevant for investigating the peel of the fruit and comparing methods of extraction for this industrial waste.

---

## Round 0.2 · accepted · Accept

Your manuscript has been accepted for publication in PeerJ

# Reviewer 1 ·

Basic reporting

no comment

Experimental design

no comment

Validity of the findings

no comment

Additional comments

The authors did a great job in this paper.

Reviewer 2 ·

Basic reporting

All the above considerations were sufficiently answered by the authors.

Experimental design

All the above considerations were sufficiently answered by the authors.

Validity of the findings

All the above considerations were sufficiently answered by the authors.

Additional comments

The authors accepted the suggestions of the review and were enlightening in their amendments. I believe the manuscript is more complete and ready to contribute to the area in which they propose to research.